# Coordination between Calcium/Calmodulin-Dependent Protein Kinase II and Neuronal Nitric Oxide Synthase in Neurons

**DOI:** 10.3390/ijms21217997

**Published:** 2020-10-27

**Authors:** Shoma Araki, Koji Osuka, Tsuyoshi Takata, Yukihiro Tsuchiya, Yasuo Watanabe

**Affiliations:** 1Department of Pharmacology, Showa Pharmaceutical University, Machida, Tokyo 194-8543, Japan; araki@ac.shoyaku.ac.jp (S.A.); tsuyoshi.takata.c2@tohoku.ac.jp (T.T.); yatsuchi@ac.shoyaku.ac.jp (Y.T.); 2Department of Neurological Surgery, Aichi Medical University, Aichi 480-1195, Japan; k_osuka@hotmail.com; 3Department of Environmental Health Sciences and Molecular Toxicology, Graduate School of Medicine, Tohoku University, Miyagi 980-8575, Japan

**Keywords:** Ca^2+^/calmodulin-dependent protein kinase II (CaMKII), nitric-oxide (NO), neuronal ischemia, phosphorylation, redox regulation

## Abstract

Ca^2+^/calmodulin (CaM)-dependent protein kinase II (CaMKII) is highly abundant in the brain and exhibits broad substrate specificity, thereby it is thought to participate in the regulation of neuronal death and survival. Nitric oxide (NO), produced by neuronal NO synthase (nNOS), is an important neurotransmitter and plays a role in neuronal activity including learning and memory processes. However, high levels of NO can contribute to excitotoxicity following a stroke and neurodegenerative disease. Aside from NO, nNOS also generates superoxide which is involved in both cell injury and signaling. CaMKII is known to activate and translocate from the cytoplasm to the post-synaptic density in response to neuronal activation where nNOS is predominantly located. Phosphorylation of nNOS at Ser847 by CaMKII decreases NO generation and increases superoxide generation. Conversely, NO-induced *S*-nitrosylation of CaMKII at Cys6 is a prominent determinant of the CaMKII inhibition in ATP competitive fashion. Thus, the “cross-talk” between CaMKII and NO/superoxide may represent important signal transduction pathways in brain. In this review, we introduce the molecular mechanism of and pathophysiological role of mutual regulation between CaMKII and nNOS in neurons.

## 1. Introduction

Ca^2+^/calmodulin (CaM)-dependent kinase II (CaMKII) is highly abundant in the brain (1 to 2% of the total brain proteins) [1,2,3]. CaMKII has a diverse intracellular localization and neuronal activation induces CaMKII to translocate from the cytoplasm to the presynaptic active zone [4] and the post-synaptic density (PSD) [5,6,7]. The mechanism of active zone accumulation of CaMKII is unclear, but the *N*-methyl-D-Aspartate (NMDA) receptor (NMDAR) subunit GluN2B (formally referred to as NR2B) has been shown to be most important for CaMKII translocation to the PSD (Figure 1a) [8,9,10,11,12]. Presynaptic CaMKII regulates neurotransmitter synthesis and release [13]. PSD-localized CaMKII undergoes autophosphorylation at Thr286 of the kinase, resulting in prolonged Ca^2+^/CaM-independent autonomous kinase activity. The autonomous CaMKII activity elicits long-lasting enhancement of synaptic plasticity, a process termed long-term potentiation (LTP), triggered by transient Ca^2+^ elevations. The increased PSD-associated CaMKII also modulates postsynaptic proteins critical for LTP maintenance [14,15,16].

CaMKII has been known to have broad substrate specificity involved in many different cellular functions. As such, CaMKII can induce either neuroprotective or neurotoxic effects after ischemia. It was shown that CaMKII autonomous activity may be a drug target for post insult neuroprotection [17,18]. However, sustained CaMKII inhibition (8–24 h) is sufficient to increase vulnerability to subsequent glutamate excitotoxic insults [19]. Isolated PSDs dephosphorylate exogenous soluble autophosphorylated CaMKII at Thr286. Protein phosphatase (PPase) 2A (PP2A) can dephosphorylate soluble CaMKII [20] but is somehow prevented from dephosphorylating the CaMKII within the PSD [21]. Thus, PP1, which is found at high levels in the PSD, might primarily be responsible for dephosphorylation of Thr286 and inactivation of PSD-associated CaMKII (Figure 1a) [22]. Thus, the reversible translocation of post-ischemic CaMKII from cytoplasm to PSDs may be important for modulating the ischemic outcome.

In brain, nitric oxide (NO) participates in numerous functions, including neurotransmission, development, and neuroprotection [23]. NO is produced in the human body from L-arginine by NO synthases (NOSs) [24]. Three major NOS isoforms have been identified, neuronal NOS (nNOS), endothelial NOS (eNOS), and inducible NOS. NO activates soluble guanylate cyclase to increase cellular cGMP levels. NO also regulates protein functions by redox-based posttranslational modification such as reversible *S*-nitrosylation, the addition of an NO to the thiol side chain of cysteine residues [25]. Besides NO, superoxide is generated by NOSs through uncoupling reactions [26,27]. NO reacts with superoxide to form a peroxynitrite, a more powerful oxidant that acts as not only a toxicant, but also a signaling molecule [28].

According to a ternary complex containing nNOS, postsynaptic density 95 kDa (PSD-95), and the NMDAR subunit GluN2B in neuron [29], the production of NO can be regulated by NMDAR (Figure 1a). NO derived from GluN2B/PSD-95/nNOS signaling plays NMDAR-dependent excitotoxicity. The peptide which disrupts the interaction between GluN2B and PSD-95/nNOS uncoupled NO production by nNOS from NMDAR-mediated excitotoxicity in cultured neurons [30] and in Alzheimer’s disease mouse models [31] and has been assessed in human ischemic stroke patients [32,33]. Meanwhile, GluN2A is considered to have a pro-survival role [34,35]. The differential roles of GluN2A- and GluN2B-containing NMDARs in neurons have been extensively reviewed [36].

nNOS exists in γ-aminobutyric acid (GABA)ergic inhibitory interneurons where CaMKII co-localizes in rat hippocampal primary neuron (Figure 1b). CaMKII phosphorylates and influences nNOS via its specific Ser847 residue [37,38] and CaMKII activity is reversibly regulated by *S-*nitrosylation via its specific Cys6 residue [39]. This review focuses on existing findings that the mutual posttranslational modifications of CaMKII and nNOS, mainly in hippocampal neurons, could be potential therapeutic signals for neuronal pathophysiology and also discusses the way that controls CaMKII activity in neurons aside from conventional inhibitors.

## 2. Regulation of nNOS by CaMKII

### 2.1. Molecular Mechanism

NO is generated by NOSs, which catalyze the conversion of L-arginine to L-citrulline and NO by transferring electrons from nicotinamide adenine dinucleotide phosphate (NADPH), via the flavin adenine dinucleotide (FAD) and flavin mononucleotide (FMN) to the heme [24,40]. When the electron transfer is impaired due to defects in the substrate L-arginine and the coenzyme tetrahydrobiopterin, NOSs utilize uncoupled electrons from NADPH to produce superoxide instead of NO. Previously, we reported that phosphorylation of nNOS at Ser847 by CaMKII and 90-kDa ribosomal S6 kinase 1 attenuates the NO synthesis activity of nNOS in vitro and in cells [37,38]. On the other hand, superoxide generation induced by nNOS is increased by phosphorylation at Ser847 (Figure 2) [28]. An electron spin resonance spectrometer coupled with the superoxide-specific spin-trapping method revealed that the phosphorylation of nNOS at Ser847 or a phosphorylation-mimic mutant of nNOS (Ser847 replaced with aspartic acid, S847D) increased uncoupling to produce a superoxide [28]. Therefore, nNOS switches NO generation to superoxide generation via the phosphorylation by CaMKII. The reason why phosphorylation at Ser847 induces nNOS uncoupling remains to be elucidated. Previously, we reported that S847D or nNOS phosphorylated at Ser847 results in lower CaM binding activity than wild type or unphosphorylated nNOS [37]. The electron transfer from the flavin domain to the heme domain by NADPH is thought to be stimulated by the binding of Ca^2+^/CaM [41,42]. Thus, the lower CaM binding ability of phosphorylation of nNOS at Ser847 might be involved in the uncoupling reaction. In this context, nNOS is also phosphorylated at Ser741, located in the CaM-binding sequence, by CaMKI (an another CaMK family), leading to reduction in CaM binding and in NO synthesis activity in cells [43]. The effects of phosphorylation of nNOS at Ser741 on superoxide production need to be elucidated. PSD-95 mediates co-localization of CaMKII and nNOS, which could be important in phosphorylation at Ser847 of nNOS by CaMKII [44].

Okadaic acid-sensitive PPases are involved in dephosphorylation of nNOS at Ser847 and reversible activation of the enzyme in NG108-15 neuronal cells [38]. Recombinant nNOS phosphorylated by CaMKII at Ser847 was dephosphorylated with brain extracts using as the source of PPases. In addition, this dephosphorylation of nNOS at Ser847 was canceled when brain extracts were co-treated with calyculin A or okadaic acid (an inhibitor of PP1 and 2A) but neither cyclosporin A (an inhibitor of calcineurin) nor protein phosphatase inhibitor-2 (an inhibitor of PP1) [45]. Furthermore, the brain-extract-mediated dephosphorylation was inhibited by calyculin A and okadaic acid with similar sensitivity. Both calyculin A and okadaic acid are reported to inhibit PP2A- but not PP1-activity with a similar potency [46]. Thus, PP2A is identified as a major PPase involved in the dephosphorylation of nNOS at Ser847, responsible for reversible activation of the enzyme in neuronal cells.

### 2.2. Experimental Models

There have been several pathophysiological roles observed to be phosphorylated nNOS at Ser847 in cells and in vivo (Table 1).

#### 2.2.1. Cultured Cells

Glutamate-induced nNOS signal is inhibited by the treatment calyculin A and okadaic acid with the similar sensitivity in mouse neostriatal slices, suggesting that nNOS at Ser847 is mainly dephosphorylated by PP2A [47]. CaMKII may efficiently phosphorylate nNOS as it can attenuate PP2A activity through the phosphorylation of its regulatory subunit [20,50]. Mouse hippocampal HT22 cells stimulated with CoCl_2_, which mimics the hypoxic conditions, up-regulate the expression of PP2A and down-regulate phosphorylation of CaMKII at most likely Thr286 and nNOS at Ser847 and a significantly decrease cell viability. Treatment of propofol ameliorate CoCl_2_/hypoxia induced cell damage via phosphorylation of nNOS at Ser847 by CaMKII [48]. Physiological concentrations (5 µM) of glutamate stimulation induce time-dependent phosphorylation at Ser847 of nNOS by CaMKII. Pathological high glutamate (100–500 µM) activates PPase and decreases the level of the phosphorylation of nNOS at Ser847, leading the overproduction of NO during pathogenesis in rat hippocampal neurons [49]. We reported that CaMKII phosphorylates not only Ser847 but also Ser1412 in cultured cells [51], which induce activation of nNOS by reducing the enzyme’s Ca^2+^ requirement and facilitating electron transfer [52]. The study of NO and superoxide generation activity of nNOS phosphorylated at not only Ser847 but also Ser1412 needs to be determined.

CaMKII is thought to regulate the neuroprotective pathway via phosphorylation at Ser847 of nNOS. In fact, in the cerebellar granule neurons, nicotine increases intracellular Ca^2+^ via nicotinic acetylcholine receptor and activates CaMKII leading to phosphorylation of nNOS at Ser847. nNOS phosphorylated at Ser847 generates superoxide that reacts with NO to form peroxynitrite. Peroxynitrite reacts with cGMP to form 8-nitro cGMP which induces protein *S*-guanylation of redox sensor protein Keap1 which results in expression of cytoprotective protein such as heme oxygenase-1 [28,53]. Thus, phosphorylation of nNOS at Ser847 can change the ratio of NO to superoxide to generate peroxynitrite and play a role not only in neurotoxic effects by itself but also neuroprotective effects against neuronal damage (Figure 3).

**Table 1 ijms-21-07997-t001:** Pathophysiological role of phospho-nNOS at Ser847 in neurons.

Tissue/Cell Type	Method	Outcome	Reference
Mouse neostriatal slices	Treatment with calyculin A or okadaic ocid	Inhibition of glutamate/NO signal	[47]
Rat hippocampal neurons	Treatment with glutamate (low/high concentration)	Inhibition of NO production	[49]
Mouse hippocampal HT22 cells	Treatment of cells exposed CoCl_2_ with propofol	Neuroprotective effects	[48]
CGCs^1^ cells, PC12 cells (nNOS expressed)	Treatment with nicotine	Neuroprotective effects	[28]
Rat hippocampus	Ischemia/reperfusion	Neuroprotective effects	[54]
Rat hippocampus	SAH (increasing ICP^2^)	Neuroprotective effects	[55]
Rat hippocampus	Treatment of GABA^3^ agonists	Neuroprotective effects	[56]
Rat hippocampus	Transient ischemic preconditionning	Neuroprotective effects	[57]
Rat hippocampus	Ischemia/reperfusion	Neuroprotective effects	[58]
Rat hippocampus	Ischemia/reperfusion	Neuroprotective effects	[59]
Mouse hippocampus	Hypothermia	Neuroprotective effects	[60]
Hamster suprachiasmatic nuclei	Diumal circadian	Photic circadian entrainment	[61]
Mouse spinal cords	Peripheral nerve injury	Neuroprotective effects	[62]
Rat supraoptic/paraventricular nuclei	Treatment of prolactin	Inhibition of vasopressin/oxytocin secretion	[63]
Mouse nucleus intermediolateralis	Spinal cord injury	Autonomic failure	[64]
Mouse periaqueductal grey matter	Treatment of morphine	Analgesic tolerance	[65]
Rat cerebellum	Hyperammonemia	Inhibition of NO production	[66]

^1^ CGCs, cerebellar granule cells; ^2^ ICP, intracranial pressure; ^3^ GABA, γ-aminobutyric acid.

#### 2.2.2. Animal Models

Under pathological conditions, such as brain ischemia, nNOS activation might be overstimulated and thus form excess NO, which triggers many downstream neurotoxic cascades form excess NO, which triggers many downstream neurotoxic cascades. Several reports have been made on the therapeutic effect of CaMKII phosphorylation of nNOS in the ischemic condition, which indicates neuroprotective effects of phosphorylation at Ser847 of nNOS by CaMKII in culture cell and animal models. Activation of nNOS plays a crucial role in neuronal injury after cerebral ischemia/hypoxia and treatment of NOS inhibitor (L-NAME, L-NNA) protected against cerebral damage [68]. In primary cortical neuronal cultures from nNOS null mice, neurotoxicity induced by hypoxia/glucose deprivation or treatment of glutamate is significantly attenuated [69]. Transient cerebral ischemia induces increased intracellular Ca^2+^ via many ways such as increase in extracellular glutamate, opening of ion channel and more [70]. With the increase in intracellular Ca^2+^ by transient ischemia, nNOS is activated and NO production is enhanced. NO induces direct *S*-nitrosylation of several proteins that are involved in neurotoxicity [71,72]. NO reacts with the superoxide to form peroxynitrite that can cause lipid peroxidation, protein nitration and direct DNA damage leading to cell death [67].

We reported for the first time that ischemia/reperfusion causes phosphorylation of nNOS at Ser847 which is inhibited by KN-93 [73] (an inhibitor of CaMKII) at nonpyramidal interneurons in the rat hippocampal CA1 but not in the cortex region [54]. The therapeutic hypothermia has been shown to improve neurologic outcomes after perinatal cerebral hypoxia/ischemia [74]. The complex of CaMKII/nNOS efficiently proceeds CaMKII-induced phosphorylation of nNOS in brain. Focal cerebral ischemia induces dissociation between CaMKII and nNOS in the CA1 hippocampus but not in the cerebral cortex under normothermia in mice. However, hypothermia maintains the association between CaMKII and nNOS in CA1 hippocampus after focal cerebral ischemia [60]. These results strongly suggest that phosphorylation at Ser847 of nNOS by CaMKII shows the therapeutic effects of hypothermia in the CA1 hippocampus. Furthermore, immediately after subarachnoid hemorrhage, an intracranial pressure is increased and induces transient cerebral ischemia, which promotes colocalization of CaMKII and nNOS, and phosphorylation at Ser847 of nNOS by CaMKII in rat hippocampus. Thus, CaMKII may improve early brain injury via phosphorylation at Ser847 of NOS during increasing intracranial pressure either with or without subarachnoid hemorrhage [55]. GABA agonists exert neuroprotective effects against excitotoxicity induced glutamate after global ischemia. The neuroprotective mechanism of GABA agonists involves increased binding of PSD-95 to nNOS and associated phosphorylation at Ser847 of nNOS [56]. Ischemic preconditioning, which is a brief, non-lethal ischemic event can ameliorate brain injury induced by subsequent severe ischemia/reperfusion [75]. Ischemic preconditioning can ameliorate ischemia/reperfusion in brain. The phosphorylation at Ser847 of nNOS by CaMKII contributes the mechanisms of neuroprotective effect of preconditioning. Ischemic preconditioning induces up regulation of phosphorylation at Ser847 of nNOS and Thr286 of CaMKII. Treatment of KN-62 [76] (an inhibitor of CaMKII) spoils the neuroprotective effect of ischemic preconditioning [57].

There are some reports on the activation of NO signals through the dephosphorylation of nNOS at Ser847, suggesting their physiological and pathological importance. In terms of circadian rhythms, phosphorylation at Ser847 of nNOS and phosphorylation at Thr286 of CaMKII are observed during the day or subjective day in the hamster suprachiasmatic nuclei. It suggests a direct pathway between both enzymes which is at least partially responsible of photic circadian entrainment by phospho- and dephospho-nNOS at Ser847 in the suprachiasmatic nuclei [61]. In the mice’s neuropathic pain in a chronic constriction injury model, D-serine induces Ca^2+^ influx via NMDAR, which evokes dephosphorylation at Ser847 of nNOS by activating calcineurin and promotes NO synthesis [62]. Prolactin induces dephosphorylation at Ser847 of nNOS and promotes vasopressin and oxytocin secretion. The phosphorylation at Ser847 of nNOS in the rat hyperprolactinemia model is lower than the control rat [63].

In contrast, there are some reports showing the pathological significance of phosphorylation at Ser847 of nNOS. In the mice models, phosphorylation at Ser847 of nNOS is significantly increased from 3 h after spinal cord injury to peak at 24 h and thereafter gradually decreased. Both nNOS phosphorylated at Ser847 and CaMKII phosphorylated at Thr286 are co-localized in the nucleus intermediolateralis after spinal cord injury. NO is thought to play a role as a neurotransmitter in autonomic preganglionic neurons. Thus, phosphorylation of nNOS at the Ser847 signaling pathway might be involved in the autonomic failure which occurs immediately after spinal cord injury [64]. In periaqueductal grey, interaction between the mu-opioid receptor and NMDAR/CaMKII-phosphorylation of nNOS at the Ser847 pathway supports the development of analgesic tolerance to morphine [65]. Phosphorylation at Ser847 of nNOS by CaMKII may promote the impairment of learning and memory. In the hyperammonemia model, CaMKII is activated, leading to increased phosphorylation of Ser847, which leads to inhibition of the NO-cGMP pathway in cerebellum [66].

## 3. Regulation of CaMKII by nNOS

### 3.1. Molecular Mechanisms

As mentioned above, CaMKII regulates NO/superoxide signals via phosphorylation of nNOS at Ser847. Conversely, CaMKII is regulated by NO via *S*-nitrosylation of active cysteine thiol. We reported that NO inhibits CaMKII activity via *S*-nitrosylation at Cys6 and/or Cys30 [39]. The *S*-nitrosylation of CaMKII exhibits ATP competitive inhibition of the enzyme activity [77]. Interestingly, Cys6 is not located at the ATP binding region of CaMKII, but modification of Cys6 reduces affinity of ATP binding to CaMKII. The result strongly suggests that *S*-nitrosylation at Cys6 mainly disturbs ATP binding to CaMKII but not that at Cys30 located in canonical ATP binding region. Cys6 is conserved in all CaMKII genes α, 𝛽, γ, and 𝛿 [78]. Thus, Cys6 plays important roles in regulation of CaMKII activity via *S*-nitrosylation. It is well established that autophosphorylation of CaMKII at Thr286 can generate a largely Ca^2+^-independent autonomous activity. Not only Ca^2+^/CaM-dependent but also autonomous activity is inhibited by NO. Thus, the binding of Ca^2+^/CaM to CaMKII is not essential for NO-induced inactivation (Figure 4).

On the other hand, NO-induced autonomous activity of CaMKII occurred by *S*-nitrosylation at Cys280/Cys289 near autophosphorylation site at Thr286 [79]. An initial Ca^2+^/CaM-stimulation is required to induce the CaMKII autonomy both in the autophosphorylation at Thr286 and *S*-nitrosylation at Cys280/Cys289. In the presence of Ca^2+^/CaM, a shorter NO exposure is sufficient to generate autonomous activity, while prolonged exposure reduces the autonomous activity either in the presence or absence of Ca^2+^/CaM (Figure 3) [79]. Therefore, it is considered that *S*-nitrosylation of Cys6 is dominant to inhibit the activity in high level and long exposure of NO in ischemic insult. Interestingly, *Caenorhabditis elegans* contains the CaMKII gene defined as unc-43 but Cys6 is not conserved to be changed to Ser5 [80] and does not encode the NOS gene [81], suggesting that *Caenorhabditis elegans* CaMKII is not able to respond to NO. Thus, the regulation of CaMKII by NO via *S*-nitrosylation at Cys6 plays an important role in mammalian cells.

### 3.2. Experimental Models

#### Cultured Cells

nNOS and CaMKII are located with PSD-95 and mediate glutamate-induced Ca^2+^ signaling. We reported that Ca^2+^ influx stimulated by glutamate induces *S*-nitrosylation at Cys6 of CaMKII in the hippocampal slice [39]. Under the ischemic condition, extracellular glutamate is increased, which evokes *S*-nitrosylation of CaMKII. Furthermore, intracellular ATP concentration is decreased by impairment of mitochondrial function in the ischemia [67]. Thus, it is suggested that CaMKII is likely to be inhibited by NO via *S*-nitrosylation at Cys6 in brain ischemia (Figure 4).

## 4. Mutual Regulation of CaMKII and nNOS

nNOS is *S*-nitrosylated at Cys331, which is one of the zinc-binding thiol that induces inactivation of nNOS. Cys331 of nNOS is highly *S*-nitrosylated in resting hippocampal CA1 and primary cortical neurons. Ischemia/reperfusion activates nNOS activity with denitrosylation at Cys331 of nNOS, coupling with the decrease in nNOS phosphorylation at Ser847 [58] (Table 1). Global cerebral ischemia induces increased *S*-nitrosylation of CaMKII, accompanied by a decrease in autophosphorylation at Thr286, suggesting a decrease in CaMKII activity. At the same time point, the phosphorylation at Ser847 and *S*-nitrosylation of nNOS were decreased, suggesting an increase in NOS activity [59]. Thus, denitrosylation/activation of nNOS may inhibit CaMKII activity via its *S*-nitrosylation at Cys6, leading to decreasing nNOS phosphorylation at Ser847 in ischemia/reperfusion models (Figure 3). Administration of *S-*nitrosoglutathione, a source of bioavailable NO, significantly decreased the neuronal damage in rat hippocampal CA1 caused by cerebral ischemia/reperfusion via *S*-nitrosylation and phosphorylation of nNOS at Ser847. Note that increasing phosphorylation of CaMKII at Thr286 and decreasing *S-*nitrosylation of CaMKII by exogenous NO donor exerted neuroprotective effects against cerebral ischemia/reperfusion injury [59] (Table 1). In this context, NO derived from nNOS plays excitotoxicity and CaMKII might induce neuroprotective effects in ischemia-dependent processes in brain.

## 5. Small Molecule Inhibitors of CaMKII, a Further Update

Cell-permeable small molecule inhibitors of specific protein kinases have proved to be invaluable reagents for elucidating pathophysiological role of a particular protein kinase [82]. KN-93/KN-62 are the most commonly used inhibitors for study of cellular and in vivo functions of CaMKII. They inhibit the CaMKII activity in a manner that is competitive with Ca^2+^/CaM but is non-competitive with ATP. Thus, neither KN-62 nor KN-93 inhibits Ca^2+^/CaM-independent autonomous activity of CaMKII phosphorylated at Thr286 (Figure 3). CaMKII can be activated by oxidative (Cys280/Met281, Cys289 (α), Met281/Met282, Cys273, and Cys290 (𝛽, γ, and 𝛿)) [83,84,85] and glycosylation (Ser279) [86] modification, resulting in autonomous activation. These autonomous activities of CaMKII are also not plausible to be inhibited by KN-62 and KN-93 [87]. Although KN-93 attenuated excitotoxicity only when present during the ischemic insult [88], it should also be noted that it blocks voltage-dependent K^+^ channels [87,89]. It has been shown that KN93 binds to CaM directly [90] and is known to alter Ca^2+^ currents in many cell types [91,92]. Therefore, the CaM antagonists would act as inhibitors of other Ca^2+^/CaM-dependent enzymes in vivo. In fact, a CaM inhibitor, DY-9760e, inhibits nNOS activity and NO production in the ischemia/reperfusion and permanent focal ischemia models [93].

It was shown that inhibiting autonomous CaMKII activity attenuates the neuronal cell death induced by excitotoxicity [18]. The peptide inhibitor tat-fused CN21 (TAT sequence fused with a 21 amino acid peptide), inhibiting both stimulated and autonomous CaMKII activity, is neuroprotective even when applied hours after a glutamate insult in primary cultures and in a mouse model [17]. The autonomous form of, in addition to the Ca^2+^/CaM-dependent form of, CaMKII activities is identified as the relevant drug targets for post-insult neuroprotection. The observed differing efficacies of exogenous inhibitors might be due to differences in these concentrations at the site of action. NO derived from an endogenous nNOS could be a safe and secure candidate for controlling CaMKII signal according pathophysiological condition in neurons (Figure 3). Although an endogenous NO does not modify CaMKII selectively, the CaMKII/nNOS complex through PSD-95 efficiently assembles a functional transduction NO on CaMKII in neurons (Figure 1a).

## 6. Conclusions

Since CaMKII and nNOS have very similar requirements for Ca^2+^/CaM, it is complicated why the adjacent steps in kinase and NO signaling should both require the same activator, Ca^2+^/CaM. PSD-localized Ca^2+^/CaM-independent autonomous CaMKII could be a candidate for the phosphorylation of nNOS which localize constitutively in PSDs. It has shown that CaMKII participates in both the neurotoxic [17,94] and neuroprotective [95] effects in ischemic insult, either with or without hypoxic preconditioning, which attenuates neuronal death induced by cerebral ischemia. However, several studies have demonstrated that inhibiting autonomous CaMKII activity attenuates the neuronal cell death induced by excitotoxicity [19,88]. The autonomous CaMKII was inhibited by exogeneous NO donors as Ca^2+^/CaM-dependent NOS [39]. Thus, both autonomous CaMKII and nNOS are mutual regulated and therapeutic targets for excitotoxic insults in neurons (Figure 3).

Phosphorylation and de-phosphorylation play important roles as a mode of signal transduction in pathophysiological processes. Additionally, the application of the site-specific phosphorylation antibody is powerful in the discovery of new signaling pathways activated by phosphorylation events in tissue/cells. Detection of protein *S*-nitrosylation using traditional methods like immunoprecipitation or Western blotting is not typically successful since the unstable S–N bonds can be broken during SDS-PAGE manipulation. Clarifying the mechanisms through the applications of enzymology, chemical biology, and mass spectrometry enable one to uncover the complexities of *S*-nitrosylation of CaMKII signals [96].

CaMKs play pivotal roles in regulating numerous functions in response to transient elevation of intracellular Ca^2+^. The activation of the CaMKs requires, by conventional means, binding of Ca^2+^/CaM to and phosphorylation of the enzymes. Recently, reactive sulfur species (RSS) were recognized as key messengers in the process of thiol-based redox regulation. It is also well-known that -SH groups of some proteins can undergo reversible polysulfidation that influences their catalytic activity. *S*-nitrosylation and polysulfidation have similar mechanisms and occurred at acid-base motif [97]. Aside from *S*-nitrosylation of CaMKII at Cys6, it was recently reported that CaMKII is sensitive to inhibition by RSS through polysulfidation at Cys6, being competitive with ATP [98,99]. Cys30 is also polysulfidated by RSS but C30A mutant of CaMKII is inhibited by RSS with ATP competitive fashion (our unpublished data). Autonomous activity of CaMKII is resistant to KN-93/KN-62, the most widely used inhibitors, but is inhibited by *S*-nitrosylation and polysulfidation. Thus, NO- and RSS-donors are a new generation of CaMKII therapeutics.

## Figures and Tables

**Figure 1 ijms-21-07997-f001:**
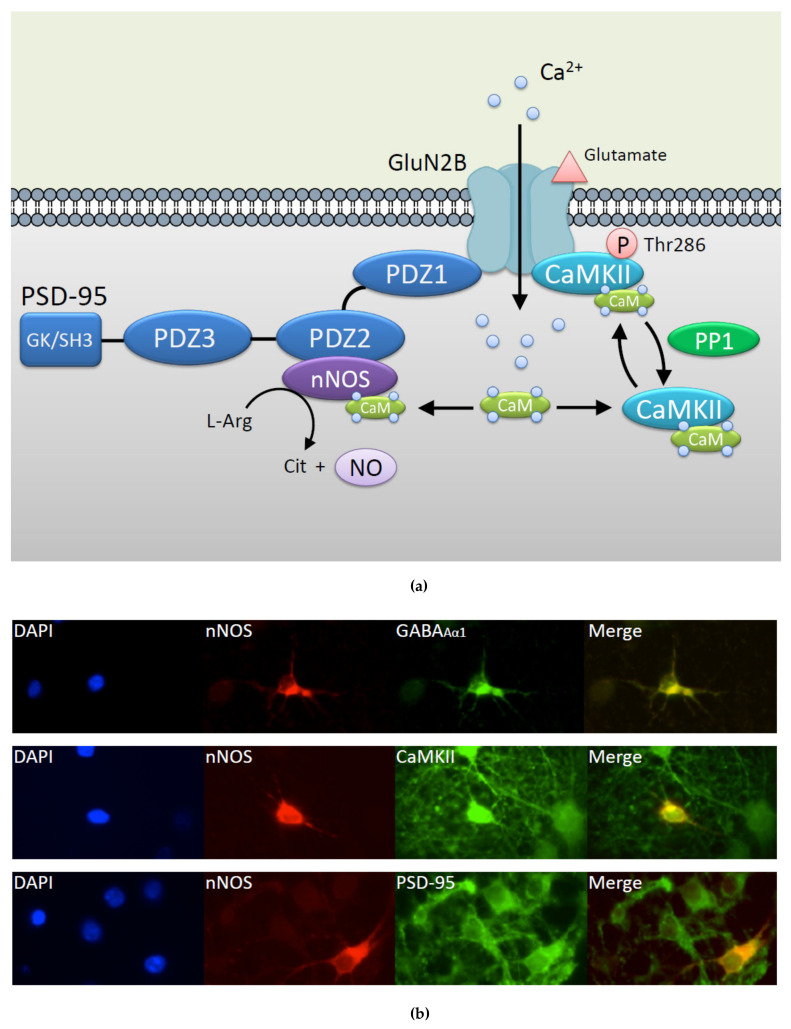
Co-localization of *N*-methyl-D-Aspartate receptor (NMDAR), post-synaptic density (PSD-95), neuronal NO synthase (nNOS), and Ca^2+^/calmodulin (CaM)-dependent protein kinase II (CaMKII) in neurons. (**a**) CaMKII and nNOS have been linked to NMDAR subunit GluN2B activation. nNOS constitutively binds to PDZ2 domain of PSD-95 associated with GluN2B. Once glutamate activates NMDAR, Ca^2+^ influx and intracellular Ca^2+^ are increased, which leads to activation of Ca^2+^/CaM binding protein including nNOS and CaMKII. Followed by Ca^2+^/CaM binding, CaMKII is immediately autophosphorylated at Thr286, which promotes interaction between CaMKII and NMDAR. protein phosphatase 1 (PP1), which is found at high levels in the PSD, might primarily be responsible for dephosphorylation of Thr286 and inactivation of PSD-associated CaMKII [22]. Under the Ca^2+^ signaling initiated by glutamate, CaMKII and nNOS are co-localized with PSD. This figure is an image of the postsynaptic hippocampal neuron where the mutual regulation of nNOS and CaMKII mainly exists. (**b**) Immunostaining micrograph images of representative rat hippocampal neurons. Panels show 4′,6-diamidino-2-phenylindole (DAPI) (blue: nuclear stain), nNOS (red), γ-aminobutyric acid (GABA)_A__α1_ (green, upper), CaMKII (green, middle), PSD-95 (green, lower). The panel in the most-right column (labelled “Merge”, shown by the yellow fluorescence) shows the overlaid image of nNOS and either GABA_Aα1_ (upper panel), CaMKII (middle panel), or PSD-95 (lower panel). Note that nNOS co-localizes with GABA_A__α1_-, CaMKII-, and PSD-95-positive neurons.

**Figure 2 ijms-21-07997-f002:**
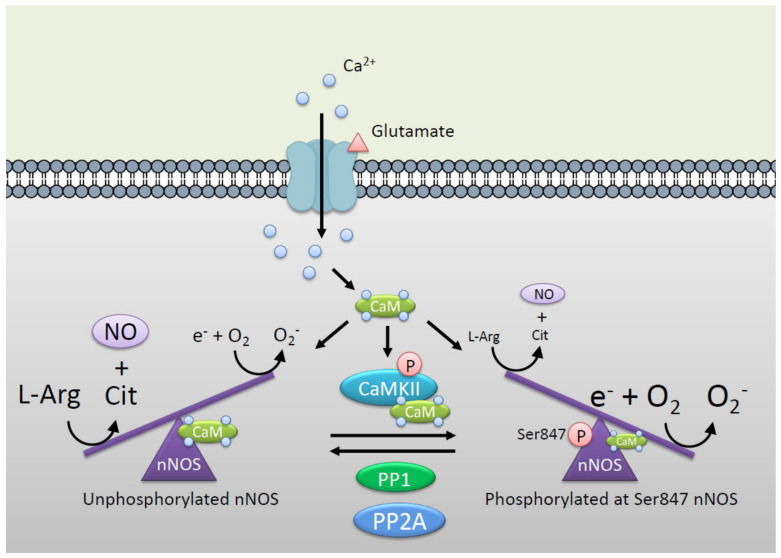
nNOS catalytic reaction is controlled by CaMKII via phosphorylation at Ser847. Unphosphorylated nNOS produces NO by conversion from L-arginine to L-citrulline via electron transfer from reductase domain to oxidase domain. Phosphorylation at Ser847 by CaMKII switches nNOS reaction from NO synthesis to superoxide (O_2_^−^) synthesis via induction of uncoupling of nNOS [28]. Protein phosphatase 2A (PP2A) [45,47,48] and PP1 [49] are identified as major protein phosphatases involved in the dephosphorylation of nNOS at Ser847, responsible for reversible activation of the enzyme in neuronal cells. This figure is an image of the postsynaptic hippocampal neuron where the mutual regulation of nNOS and CaMKII mainly exists.

**Figure 3 ijms-21-07997-f003:**
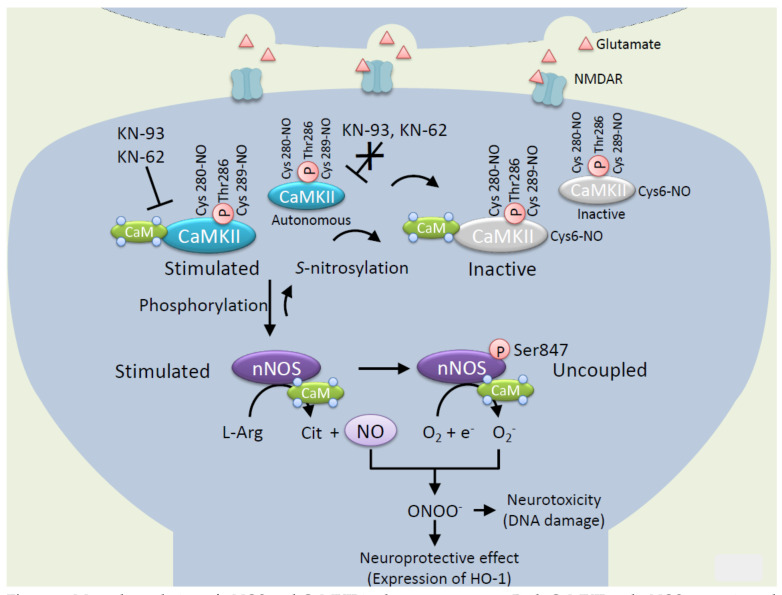
Mutual regulation of nNOS and CaMKII in the nerve systems. Both CaMKII and nNOS are activated by CaM binding (Stimulated). CaMKII is either autophosphorylated at Thr286 or *S*-nitrosylated at Cys280/Cys289 and earns Ca^2+^ independent autonomous activity. KN-93 and KN-62 generally used as CaMKII inhibitors inhibit CaM binding of CaMKII and thereby autonomous activity of CaMKII is not inhibited by KN-93 and KN-62. nNOS-derived NO inhibits CaMKII activity and both Ca^2+^ dependent and Ca^2+^ independent autonomous activity via *S*-nitrosylation at Cys6 with ATP competitive fashion [39]. On the other hand, CaMKII phosphorylates at Ser847 of nNOS which decrease NO synthesis and increase superoxide (O_2_^–^) synthesis. O_2_^–^ reacts with NO to form peroxynitrite which indicates neurotoxic effects with oxidation of DNA [67]. Peroxynitrite also works as a signal molecule when it induces expression of cytoprotective proteins such as heme oxygenase-1 (HO-1) [28]. Ca^2+^ signals orchestrate NO and superoxide and phosphorylation by interaction between CaMKII and nNOS in the central nervous system. This figure is an image of the postsynaptic hippocampal neuron where the mutual regulation of nNOS and CaMKII mainly exists.

**Figure 4 ijms-21-07997-f004:**
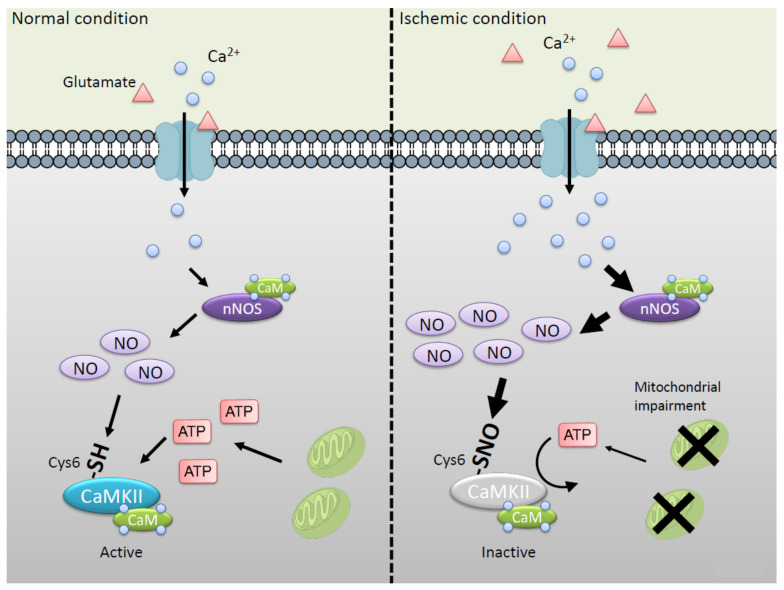
Inactivation of CaMKII via *S*-nitrosylation at Cys6 in the ischemic condition. In the normal condition (left panel), CaMKII and nNOS are controlled by glutamate signaling via NMDAR. Note that high levels of glutamate in the extracellular space are well known to occur early in the onset of ischemia, leading to the enhanced Ca^2+^ entry and the neuronal injury (right panel) [70]. At the same time, intracellular ATP concentration is decreased by mitochondrial impairment [67]. NO inhibits CaMKII activity via *S-*nitrosylation at Cys6 with ATP competitive fashion. *S*-nitrosylated CaMKII tends to be inhibited under the condition of low ATP concentration. This figure is an image of the postsynaptic hippocampal neuron where the mutual regulation of nNOS and CaMKII mainly exists.

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
