# Peer review of "Coordination between Calcium/Calmodulin-Dependent Protein Kinase II and Neuronal Nitric Oxide Synthase in Neurons"

_ijms, 2020, doi:10.3390/ijms21217997_

Round 1
Reviewer 1 Report
The review article entitled “Coordination between calcium/calmodulin-2 dependent protein kinase II and neuronal nitric oxide 3 synthase in neuron” has some questions should be solved before making a decision.
Major questions
- Does this pathway exist in the whole brain or specific region?
- Authors should mention the brain area in each figure if it is necessary.
Author Response
Dear Dr.-Reviewer 1
Thank you very much for your report regarding our manuscript entitled " Coordination between calcium/calmodulin-dependent protein kinase II and neuronal nitric oxide synthase in neurons " We appreciate the insightful suggestions provided from you. Our manuscript has been extensively revised according your suggestions. The main concerns of the reviewers are addressed below:
-Reviewer 1
The review article entitled “Coordination between calcium/calmodulin-2 dependent protein kinase II and neuronal nitric oxide 3 synthase in neuron” has some questions should be solved before making a decision.
Point 1: Does this pathway exist in the whole brain or specific region?
Response 1: This pathway exists mainly in hippocampus and also in spinal cords, cerebellum, suprachiasmatic nuclei, supraoptic/paraventricular nuclei, nucleus intermediolateralis, and periaqueductal grey matter as indicated in Table 1. As reviewer’s comment, we added the sentence as followed, “This review focuses on existing findings that the mutual posttranslational modifications of CaMKII and nNOS mainly in hippocampal neurons, could be potential therapeutic signals for neuronal pathophysiology and also discusses the way which controls CaMKII activity in neurons aside from conventional inhibitors” (see lane 76 , highlighted in red).
- Authors should mention the brain area in each figure if it is necessary.
Response 2: As described above, the pathway exists mainly in hippocampus. As reviewer’s comment, we added the following sentence in each figure legend. “This figure is an image of the postsynaptic hippocampal neuron where the mutual regulation of nNOS and CaMKII mainly exists in” (see the legends of figure 1-4, highlighted in red)
Additionally, the manuscript has been revised accordingly (highlighted in red). We hope that these changes along with the additional data address the reviewers’ concerns.
We hope that the revised manuscript is now suitable for publication in an IJMS review article on a Special Issue on "Role of CaM kinase II in Nervous System"
Sincerely yours,
Yasuo Watanabe, MD, Ph.D.
Professor
Pharmacology
Showa Pharmaceutical University
3-3165, Higashi-tamagawa Gakuen
Machida, Tokyo 194-8543, Japan
Tel/Fax:+81-42-721-1549
E-mail:yasuwata@ac.shoyaku.ac.jp

Reviewer 2 Report
Summary
The authors provide a nice review of CaMKII and nitric oxide (NO) in neurons.
Major points:
-Figure 1b legend is confusing – the CaMKII and PSD96 images are green/yellow and yellow however in the legend it refers to different colors, this needs to be clarified or revised. Also CaMKII should not be hyphenated in the image as it is not hyphenated anywhere else.
-Section on inhibitors. It’s worth mentioning that there are no structures of either inhibitor mentioned bound to CaMKII, so we do not fully understand their function. Additionally, it’s been shown that KN93 binds to calmodulin directly and is known to alter calcium currents in many cell types, this should be clarified.
Minor points:
-title heading for section 2 is the same as section 3
-Figure 1a -> would be nice to label the subunits of NMDA receptor since you refer to these in the text
-Figure 3 -> there are plenty of CaMKII structures available, not sure why the CaMKII cartoon here looks different from the others? If it’s to indicate inactive, I suggest keeping the same shape and changing the color.
-Table 1 is a very nice reference tool
-Lines 73-75, would be easier to follow this if you include the specific residue numbers
-Line 142: change indicate to result in
-lines 143-145: awkward, needs to be rephrased
-line 147: remove ‘aside from CaMKII’
-154-156: awkward, needs to be rephrased
-165: not sure you mean ‘afford’ here
-167: conditions
-170-172: this is unclear – inhibition of phosphatase leads to less phosphorylation? Needs explanation if this statement is correct
-174: ‘study of enzyme activity’ is unclear – what exactly needs to be measured?
-177: reword sentence to start with ‘Pathological concentrations of glutamate induces excitotoxicity, which activates phosphatases… ‘ (missing some plurals and descriptors here)
-182: change ‘that… ’ to ‘leading to phosphorylation of nNOS at Ser847’
-193: conditions, indicates
-204: this sentence is confusing, do you mean causes KN93 sensitivity?
-213: also confusing, what do you mean by ‘involved in therapeutic effects’?
-218: GABA agonists exert, or A GABA agonist exerts
-222: would be good to define preconditioning earlier
-247: CaMKII is activated, leading to increased phosphorylation of Ser847…
-251: remove ‘a’
-258: conserved in all CaMKII genes (or paralogs) alpha, beta etc are not isoforms
-261: not sure what you mean by 286 phospho camkii is similarly inhibited by Ca/CaM dependent enzyme?
-263: induced
-279, 283: nitrosylation spelling
Author Response
Dear Dr.-Reviewer 2
Thank you very much for your reports regarding our manuscript entitled " Coordination between calcium/calmodulin-dependent protein kinase II and neuronal nitric oxide synthase in neurons " We appreciate the insightful suggestions provided from you. Our manuscript has been extensively revised according your suggestions. The main concerns of the reviewers are addressed below:
-Reviewer 2
The authors provide a nice review of CaMKII and nitric oxide (NO) in neurons.
- Figure 1b legend is confusing – the CaMKII and PSD96 images are green/yellow and yellow however in the legend it refers to different colors, this needs to be clarified or revised. Also CaMKII should not be hyphenated in the image as it is not hyphenated anywhere else.
Response 1: In figure 1b, in the upper panel, the cells were double-immunostained with anti-nNOS (shown by the red fluorescence: nNOS) and anti-GABAAα1 (shown by the green fluorescence: GABAAα1). The panel in the most right column (labelled ‘Merge’) shows the overlaid image (shown by the yellow fluorescence). In the middle panel, the cells were also double-immunostained with anti-nNOS (shown by the red fluorescence: nNOS) and anti-CaMKII (shown by the green fluorescence: CaMKII) in the middle panel. The panel in the most right column (labelled ‘Merge’) shows the overlaid image (shown by the yellow fluorescence). In the lower panel, the cells were double-immunostained with anti-nNOS (shown by the red fluorescence: nNOS) and anti-PSD-95 (shown by the green fluorescence: PSD-95) in the middle panel. The panel in the most right column (labelled ‘Merge’) shows the overlaid image (shown by the yellow fluorescence).
As reviewer’s comment, we have revised the sentence from “and the colocalization of nNOS and either GABAA?1 (yellow, upper), CaMKII (yellow, middle), or PSD95 (yellow, lower).” to “The panel in the most right column (labelled ‘Merge’, shown by the yellow fluorescence) shows the overlaid image of nNOS and either GABAAα1 (upper panel), CaMKII (middle panel), or PSD-95 (lower panel) (see figure 1b legend, highlighted in red).
As reviewer’s comment, we have changed from CaM-KII to CaMKII and from PSD95 to PSD-95 (see figure 1b).
- Section on inhibitors. It’s worth mentioning that there are no structures of either inhibitor mentioned bound to CaMKII, so we do not fully understand their function. Additionally, it’s been shown that KN93 binds to calmodulin directly and is known to alter calcium currents in many cell types, this should be clarified.
Response 2: As reviewer’s comment, we have mentioned as follows, “It has been shown that KN93 binds to CaM directly [90] and is known to alter Ca2+ currents in many cell types [91, 92]. Therefore, the CaM antagonists would act as inhibitors of other Ca2+/CaM-dependent enzymes in vivo. In fact, a CaM inhibitor, DY-9760e, inhibits nNOS activity and NO production in the ischemia/reperfusion and permanent focal ischemia models [93]. (see lanes 323-327, highlighted in red)
Minor points:
-title heading for section 2 is the same as section 3
Response: As reviewer’s comment, we have changed from “3 Regulation of nNOS by CaMKII” to “3 Regulation of CaMKII by nNOS” (see lane 261, highlighted in red)
-Figure 1a -> would be nice to label the subunits of NMDA receptor since you refer to these in the text
Response: As reviewer’s comment, we have labeled the subunits of NMDA receptor as GluN2B (see figure 1a).
-Figure 3 -> there are plenty of CaMKII structures available, not sure why the CaMKII cartoon here looks different from the others? If it’s to indicate inactive, I suggest keeping the same shape and changing the color.
Response: As reviewer’s comment, we have kept the same shape and changed the color of the CaMKII cartoon in figure 3 (see figure 3 and 4).
-Table 1 is a very nice reference tool.
Response: We appreciate the comments.
-Lines 73-75, would be easier to follow this if you include the specific residue numbers
Response: As reviewer’s comment, we have included the specific residue numbers as Ser847 residue and Cys6 (see lanes 74 and 75, highlighted in red).
-Line 142: change indicate to result in
Response: As reviewer’s comment, we have changed indicate to result in (see lane 153, highlighted in red).
-lines 143-145: awkward, needs to be rephrased
Response: As reviewer’s comment, we have rephrased as “The electron transfer from the flavin domain to the heme domain by NADPH is thought to be stimulated by the binding of Ca2+/CaM [41, 42]. Thus, the lower CaM binding ability of phosphorylation of nNOS at Ser847 might be involved in the uncoupling reaction.” (see lanes 154-157, highlighted in red)
-line 147: remove ‘aside from CaMKII’
Response: As reviewer’s comment, we have removed ‘aside from CaMKII’ (see lane 158, highlighted in red)
-154-156: awkward, needs to be rephrased
Response: As reviewer’s comment, we have rephrased as “And this dephosphorylation of nNOS at Ser847 was canceled when brain extracts were co-treated with calyculin A or okadaic acid (an inhibitor of PP1 and 2A) but neither cyclosporin A (an inhibitor of calcineurin) nor protein phosphatase inhibitor-2 (an inhibitor of PP1) [45]. Furthermore, the brain extracts-mediated dephosphorylation was inhibited by calyculin A and okadaic acid with the similar sensitivity. Both calyculin A and okadaic acid are reported to inhibit PP2A- but not PP1- activity with a similar potency [46].” (see lanes 165-170, highlighted in red)
-165: not sure you mean ‘afford’ here
Response: As reviewer’s comment, we have changed ‘afford’ to ‘efficiently’ (see lanes 178, highlighted in red)
-167: conditions
Response: As reviewer’s comment, we have changed ‘condition’ to ‘conditions’ (see lanes 180, highlighted in red)
-170-172: this is unclear – inhibition of phosphatase leads to less phosphorylation? Needs explanation if this statement is correct
Response: What is “activation of phosphatase leads to less phosphorylation”. As reviewer’s comment, we have rephrased as “Pathological high glutamate (100-500 µM) activates PPase and decreases the level of the phosphorylation of nNOS at Ser847, leading the overproduction of NO during pathogenesis in rat hippocampal neurons [50].” (see the response to -177: and lanes 185-187 in the text, highlighted in red)
-174: ‘study of enzyme activity’ is unclear – what exactly needs to be measured?
Response: As reviewer’s comment, we rephrased “The study of NO and superoxide generation activity of nNOS phosphorylated at not only Ser847 but also Ser1412 needs to be determined.” (see lane 189, highlighted in red)
-177: reword sentence to start with ‘Pathological concentrations of glutamate induces excitotoxicity, which activates phosphatases… ‘ (missing some plurals and descriptors here)
Response: As reviewer’s comment, we have responded in the above (see the response to -170-172)
-182: change ‘that… ’ to ‘leading to phosphorylation of nNOS at Ser847’
Response: As reviewer’s comment, we have changed ‘that… ’ to ‘leading to phosphorylation of nNOS at Ser847’ (see lane 193, highlighted in red)
-193: conditions, indicates
Response: As reviewer’s comment, we have changed ‘condition, which indicate’ to ‘condition, indicates’ (see lane 204, highlighted in red)
-204: this sentence is confusing, do you mean causes KN93 sensitivity?
Response: As reviewer’s comment, we have rephrased “We reported for the first time that ischemia/reperfusion causes phosphorylation of nNOS at Ser847 which is inhibited by KN-93 [60] (an inhibitor of CaMKII) at nonpyramidal interneurons in the rat hippocampal CA1 but not in the cortex region [61].” (see lanes 215-217, highlighted in red)
-213: also confusing, what do you mean by ‘involved in therapeutic effects’?
Response: As reviewer’s comment, we have rephrased “These results strongly suggest that phosphorylation at Ser847 of nNOS by CaMKII shows the therapeutic effects of hypothermia in the CA1 hippocampus.” (see lane 223, highlighted in red)
-218: GABA agonists exert, or A GABA agonist exerts
Response: As reviewer’s comment, we have changed to “GABA agonists exert” (see lane 229, highlighted in red)
-222: would be good to define preconditioning earlier
Response: As reviewer’s comment, we have defined preconditioning earlier as “Ischemic preconditioning, which is a brief, non-lethal ischemic event can ameliorate brain injury induced by subsequent severe ischemia/reperfusion [66].” (see lanes 231-233, highlighted in red)
-247: CaMKII is activated, leading to increased phosphorylation of Ser847…
Response: As reviewer’s comment, we have rephrased as “CaMKII is activated, leading to increased phosphorylation of Ser847” (see lanes 259-260, highlighted in red)
-251: remove ‘a’
Response: As reviewer’s comment, we have removed ‘a’ (see lane 263, highlighted in red)
-258: conserved in all CaMKII genes (or paralogs) alpha, beta etc are not isoforms
Response: As reviewer’s comment, we have changed to “conserved in all CaMKII genes ?, ?, ?, and ? (see lane 270, highlighted in red)
-261: not sure what you mean by 286 phospho camkii is similarly inhibited by Ca/CaM dependent enzyme?
Response: As reviewer’s comment, we have rephrased as “Not only Ca2+/CaM-dependent but also autonomous activity is inhibited by NO.” (see lanes 272-273, highlighted in red)
-263: induced
Response: As reviewer’s comment, we have changed from induce to induced (see lane 275, highlighted in red)
-279, 283: nitrosylation spelling
Response: As reviewer’s comment, we have correct the epelling as S-nitrosylation and S-nitrosylated (see lanes 292 and 296, highlighted in red)
Additionally, the manuscript has been revised accordingly (highlighted in red). We hope that these changes along with the additional data address the reviewers’ concerns.
We hope that the revised manuscript is now suitable for publication in an IJMS review article on a Special Issue on "Role of CaM kinase II in Nervous System"
Sincerely yours,
Yasuo Watanabe, MD, Ph.D.
Professor
Pharmacology
Showa Pharmaceutical University
3-3165, Higashi-tamagawa Gakuen
Machida, Tokyo 194-8543, Japan
Tel/Fax:+81-42-721-1549
E-mail:yasuwata@ac.shoyaku.ac.jp
